# STAT3: Versatile Functions in Non-Small Cell Lung Cancer

**DOI:** 10.3390/cancers12051107

**Published:** 2020-04-29

**Authors:** Julian Mohrherr, Iris Z. Uras, Herwig P. Moll, Emilio Casanova

**Affiliations:** 1Department of Physiology, Center of Physiology and Pharmacology & Comprehensive Cancer Center (CCC), Medical University of Vienna, AT-1090 Vienna, Austria; 2Ludwig Boltzmann Institute for Cancer Research (LBI-CR), AT-1090 Vienna, Austria; 3Department of Pharmacology, Center of Physiology and Pharmacology & Comprehensive Cancer Center (CCC), Medical University of Vienna, AT-1090 Vienna, Austria

**Keywords:** signal transducer and activator of transcription (STAT), Janus kinase (JAK), cytokines, interleukin (IL), non-small cell lung cancer (NSCLC), lung adenocarcinoma (AC), Kirsten rat sarcoma viral proto-oncogene (K-RAS), epidermal growth factor receptor (EGFR), tumor microenvironment (TME), tumor-promoting inflammation, anti-tumor immunity, clinical trials

## Abstract

Signal Transducer and Activator of Transcription 3 (STAT3) activation is frequently found in non-small cell lung cancer (NSCLC) patient samples/cell lines and STAT3 inhibition in NSCLC cell lines markedly impairs their survival. STAT3 also plays a pivotal role in driving tumor-promoting inflammation and evasion of anti-tumor immunity. Consequently, targeting STAT3 either directly or by inhibition of upstream regulators such as Interleukin-6 (IL-6) or Janus kinase 1/2 (JAK1/2) is considered as a promising treatment strategy for the management of NSCLC. In contrast, some studies also report STAT3 being a tumor suppressor in a variety of solid malignancies, including lung cancer. Here, we provide a concise overview of STAT3‘s versatile roles in NSCLC and discuss the yins and yangs of STAT3 targeting therapies.

## 1. Introduction

Lung cancer is the deadliest form of cancer. Based on the figures from the Global Cancer Incidence, Mortality and Prevalence (GLOBOCAN) database, lung cancer was estimated to account for approximately 1.8 million deaths in 2018 and 2.1 million cases were newly diagnosed globally. Thus, lung cancer is responsible for almost every fifth (18.4%) cancer death. [1] The predominant risk factor for lung cancer is tobacco smoking, accounting for approximately 80% of the cases in countries where smoking is common [2]. Histologically two major subtypes can be distinguished: small-cell lung cancer (SCLC) and non-small cell lung cancer (NSCLC), with the latter one being the dominant subtype [3,4]. NSCLC can be further subdivided into lung adenocarcinoma, lung squamous cell carcinoma and large-cell lung carcinoma [3,4]. As the most abundant drivers of NSCLC tumorigenesis, mutations in tumor suppressor genes *TP53*, *KEAP1, STK11, NF1* and in the oncogenes Kirsten Rat Sarcoma Viral Proto-Oncogene *(K-RAS)* and Epidermal growth factor receptor *(EGFR)* were identified [3,5,6,7]. In addition, chronic inflammation induced by tobacco smoke or asbestos promotes NSCLC tumorigenesis and may trigger somatic driver mutations [8,9]. Current treatment strategies for NSCLC include surgery, radiation, chemotherapy, targeted therapy or immunotherapy alone or in combination [10]. Selected patients harboring actionable driver mutations (e.g., EGFR mutations) benefit from targeted therapies based on tyrosine kinase inhibitors (TKIs) or from immune checkpoint blockers (ICB). However, approximately 80% of patients suffering from NSCLC progress to stage IV tumors, and the 5-year relative survival rate is under 20% [3,4,11,12,13]. Therapies are often plagued by resistance mechanisms which blunt the initial tumor responses to TKI or ICB therapies. In addition, to date there is no approved therapy targeting mutated K-RAS (present in 30% of NSCLC patients) [14,15,16]. Consequently, researchers focus on finding alternative druggable drivers in NSCLC to improve existing therapies or provide new ones. In this regard, Signal Transducer and Activator of Transcription 3 (STAT3) and its upstream activators Interleukin-6 (IL-6) and Janus kinase 1/2 (JAK1/2), are considered as promising targets because STAT3 is frequently activated in NSCLC and regulates key cancer hallmarks, such as cell proliferation, tumor-promoting inflammation and evasion of anti-tumor immunity [12,17,18,19]. However, STAT3 may also function as a tumor suppressor in NSCLC and other solid malignancies, depending on the tumor driver and cellular context [12,20]. Within this article, we will discuss the role of STAT3 in NSCLC regarding its tumor cell-intrinsic and extrinsic mechanisms.

## 2. Homeostatic STAT3 Signaling

STAT3 was originally described as an acute phase response factor (APRF) and identified as a key mediator of IL-6-type cytokine signaling [21,22,23,24,25,26,27]. As a core component of the JAK-STAT pathway, which consists of seven STAT family members (STAT1, STAT2, STAT3, STAT4, STAT5A, STAT5B, STAT6) and four JAKs (JAK1, JAK2, JAK3, Tyrosine kinase 2 or Tyk2), STAT3 operates as a transcription factor downstream of multiple cytokines, interferons, hormones and growth factors [28,29]. STAT3, comparable to other family members, comprises six domains: a conserved-amino-terminus, a coiled-coil domain, a DNA-binding domain, a linker domain, the Src Homology 2 (SH2) domain for receptor binding and dimerization and the C-terminal transactivation-domain (TAD) for co-factor interactions. STAT3′s tyrosine residue (Tyr^705^), which becomes phosphorylated upon activation, is located between the SH2-domain and the TAD [28,29]. 

Canonic STAT3 signaling starts with extracellular ligand binding (e.g., IL-6) to the cognate cell surface receptor (e.g., gp130/IL-6R); leading to receptor dimerization and trans-phosphorylation/activation of JAKs. Activated JAKs subsequently phosphorylate cytoplasmic receptor-tails, thereby providing docking sides for STAT(3)s. STAT3 is then activated by JAKs due to single tyrosine-residue phosphorylation on its C-terminus (Tyr^705^). Once activated, STAT3 dissociates from the receptor/kinase complex and forms homodimers (STAT3:STAT3) or heterodimers (STAT3:STAT1) via SH2-domain- interactions. STAT3-dimers translocate into the nucleus where they regulate gene transcription. Under physiological conditions, STAT3 activation is rapid and transient due to the tight negative regulation by Suppressor of Cytokine Signaling (SOCS) proteins, Protein Inhibitor of Activated STAT (PIAS) proteins and phosphatases [12,18,28,30,31,32,33]. For STAT3, SOCS3 has been identified as a primary transcriptional target, which induces negative feedback regulation by impairing JAK activity [30,34]. Further, PIAS3 prevents STAT3-DNA-target binding and phosphatases like SHP-1, SHP-2, PTP1B or T-cell PTB inhibit STAT3 activity either at JAK kinase level or directly in the nucleus (Figure 1). Disruption of negative regulation renders STAT3 constitutively active, which can induce malignant cellular transformation [30,33,35]. Other post-translational modifications such as an additional serine phosphorylation (Ser^727^), acetylation or methylation also influence the transcriptional output of STAT3 [30].

## 3. STAT3 as an Oncogene in Solid Tumors

Several landmark papers in the mid and late 1990s reported the oncogenic properties of STAT3 [18]. First, Jove and colleagues showed that STAT3 is constitutively active in SRC oncoprotein transformed cells [36]. Next, it was shown that blockage of STAT3 signaling abrogates fibroblast transformation by SRC [37,38]. The first direct evidence for STAT3 as an oncogene emerged from a study using STAT3-C, a spontaneously dimerizing and constitutively active STAT3 mutant. STAT3-C expression in fibroblast induced malignant transformation [39]. Since then, various reports substantiated STAT3′s engagement in promoting cancer hallmarks, such as tumor cell proliferation, evasion of apoptosis and angiogenesis by controlling the expression of pro-tumorigenic genes like Cyclin D1/2, c-MYC, MCL1, Survivin, BCL-X_L_, HGF, HIF-1α and VEGF [12,17,18,19,40,41]. In solid cancers that generally lack STAT3 activating somatic mutations, STAT3 activation is mainly mediated by dysregulated cytokine and growth factor receptor signaling via engagement of receptor-associated JAKs. In addition, constitutively active non-receptor tyrosine kinases (nRTKs) like SRC or ABL facilitate STAT3 activation in cancer [40,42]. 

## 4. STAT3 as an Oncogene in NSCLC

The majority of NSCLC patient samples and cell lines harbor persistent STAT3 activation and high intra-tumoral pSTAT3 levels correlate with advanced disease stage, smoking and *EGFR*-mutation status [43,44,45,46,47]. Enhanced pSTAT3 expression is also associated with induction of angiogenesis in patient tumor samples and serves as a negative predictor for patient survival [44,48]. First direct evidence for STAT3 as an oncogene in NSCLC came from loss of function studies: STAT3 inhibition with an antisense STAT3 oligonucleotide markedly reduces the survival of a NSCLC cell line [47]. Additional studies investigating STAT3 in EGFR driven lung tumorigenesis confirmed its oncogenic role in NSCLC. Expression of mutant-*EGFR* variants (e.g., L858R substitution and exon 20 insertion) in NIH3T3 cells led to increased STAT3 activation compared to NIH3T3 cells expressing wild-type *EGFR* [49]. Expression of a dominant negative STAT3 form in NIH3T3 cells hindered malignant transformation induced by *EGFR* mutation [50]. Intriguingly, STAT3 activation by mutant-*EGFR* was initially believed to be direct. Nevertheless, in vitro treatment of *EGFR*-mutated NSCLC cell lines with the first generation EGFR-TKIs erlotinib or gefitinib failed to impair pSTAT3 levels in the majority of tested cell lines, indicating that STAT3 is activated by mechanisms independent of mutant-*EGFR* in this context [45,51]. In fact, treatment of *EGFR* mutated NSCLC cell lines with erlotinib increased pSTAT3 levels, thereby promoting drug resistance. Conversely, knockdown of STAT3 by RNA interference enhanced sensitivity to erlotinib [52]. Importantly, these findings could be extended to K-RAS-driven NSCLC tumorigenesis. Treatment of *K-RAS*-mutated NSCLC cell lines with a MEK inhibitor elevated pSTAT3 levels and induced drug resistance [52]. Another mechanism underlying STAT3 activation in NSCLC involves gp130/JAK signaling. NSCLC cell lines with *EGFR* and *K-RAS* mutations secrete cytokines such as IL-6. This leads to autocrine JAK-dependent STAT3 activation via gp130 engagement, thereby promoting tumor cell survival [53,54]. In this context, genetic knockdown of IL-6 or JAK1/2 decreased tumor cell growth in vivo and/or in vitro [53,55,56]. Further, single nucleotide polymorphisms (SNPs) in cytokine receptors may be critical for STAT3 activation in NSCLC. Indeed, expression of a growth hormone receptor SNP variant (*GHR*P495T) in normal human lung epithelial cells led upon growth hormone stimulation to increased STAT3 activation and cell proliferation compared to cells expressing the wild-type receptor [57].

Collectively, these studies provided evidence that increased STAT3 activation within tumor cells promotes tumor development while increased STAT3 activation within normal lung epithelial cells induces cellular transformation. However, these studies did not address the role of STAT3 in shaping the tumor microenvironment (TME).

## 5. Inflammation and Anti-Tumor Immunity: A Central Role for STAT3 in NSCLC

The TME in the lung, comparable to the TME of other solid cancers, consists of cellular and soluble factors (e.g., cytokines or growth factors), some of them exhibiting either pro- or anti-tumorigenic properties [58,59,60]. M2-polarized tumor-associated macrophages (TAMs), myeloid derived suppressor (MDSC), regulatory T (Treg), type 17 CD4^+^ (Th17), type 2 CD4^+^ (Th2), B cells or cytokines such as IL-6, IL-10, IL-17, IL-23, are generally reported to be pro-tumorigenic. By contrast, M1-polarized macrophages, cytotoxic CD8^+^, type 1 CD4^+^ (Th1), dendritic (DC), natural killer (NK) cells or pro-inflammatory cytokines such as IL-12 are considered to be anti-tumorigenic. Skewing the balance to more pro-tumorigenic factors leads to tumor outgrowth through tumor-promoting inflammation and impaired anti-tumor immunity. STAT3 signaling within tumor and immune cells shapes the TME; the vast majority of lung cancer studies report that STAT3 drives tumor-promoting inflammation while impairing anti-tumor immunity [18,19,58,59,60,61,62,63,64,65,66]. An initial study by Li et al. showed that STAT3-C overexpression in alveolar type II cells leads to severe pulmonary inflammation, ultimately resulting in lung adenocarcinoma formation in mice. A main characteristic of the lung TME within their model was the high abundance of TAMs, B-cells and CD3^+^ lymphocytes [67]. Further, lung epithelium-specific deletion of STAT3 increased NK cell immunity, thereby impairing tumor growth in a carcinogen-induced lung cancer mouse model. In vitro studies corroborated these data for human lung tumorigenesis: STAT3 knockdown in human NSCLC cell lines reduced their MHC class I expression, thereby increasing their susceptibility to NK cell-mediated cytotoxicity [68]. Tumor cell-intrinsic STAT3 also modulates cytotoxic T-cell response by regulating expression of immune checkpoint molecules such as Programmed cell death 1 ligand 1 (PD-L1). Interferon gamma (IFNγ) stimulation of NSCLC cells induced PD-L1 expression, whereas blockage of STAT3 activity abrogated it, thereby rendering cells more susceptible to cytotoxic T-cell mediated killing [69,70,71]. In addition, STAT3 gene silencing restored NSCLC cell susceptibility to cytotoxic T-cell mediated killing under hypoxic conditions, which frequently occur in the TME and impair cytotoxic T-cell function [72]. Notably, tumor cell-intrinsic STAT3 drives the expression of pro-tumorigenic cytokines (e.g., IL-6) and growth factors (e.g., VEGF), thus enhancing the recruitment of tumor-promoting myeloid cells such as TAMs or MDSCs to the TME. These cells secrete pro-tumorigenic cytokines and growth factors to amplify STAT3 activation within the tumor cells, resulting in tumor growth [44,73,74,75,76,77,78,79,80,81,82,83]. The role of STAT3 signaling in immune cells and its effects on tumor-promoting inflammation and anti-tumor immunity was also addressed in lung cancer. For instance, Zhou et al. showed that lysozyme M-Cre mediated myeloid STAT3 deletion abrogates urethane-induced lung carcinogenesis in mice by decreasing levels of pro-tumorigenic M2 macrophages, MDSCs, and Tregs, while increasing levels of anti-tumorigenic M1 macrophages, cytotoxic CD8^+^ and CD4^+^ Th1 cells in the mouse lungs [84]. Studies using pharmacological approaches, siRNA technology, *Mx1*-Cre, *CD4*-Cre or *Ncr1*-iCreTg mice to abrogate STAT3 signaling in immune cells further confirmed that STAT3 impairs anti-tumor immunity. More precisely, STAT3 activation within NK and CD8^+^ T cells hampers their cytotoxic activity, while STAT3 activation in DC cells impairs their maturation and ability to present antigens. Accumulation and differentiation of Tregs within the TME is also STAT3 dependent (Figure 2) [85,86,87,88,89,90]. Altogether, data from these studies indicate that STAT3‘s function extends far beyond the tumor cell and that blockade of STAT3 signaling might be a promising option to treat NSCLC patients. 

## 6. Targeting STAT3 Signaling in NSCLC: Past and Future

Although direct targeting of STAT3 is notoriously difficult, promising new targeting approaches continuously emerge. For instance, Bai et al. recently showed that SD-36, a highly selective small-molecule degrader of STAT3, potently inhibits growth of a leukemia and lymphoma cell line(s) in vitro and induces durable tumor regression in vivo. Hence, further investigations, specifically in NSCLC, are highly warranted [91,92,93]. Apart from SD-36, AZD9150, a next generation antisense oligonucleotide inhibitor of STAT3, may hold great promise for the treatment of NSCLC. Notably, AZD9150 potently blocked growth of a xenografted NSCLC cell line, and clinical trials in NSCLC patients have already been launched [94]. Nevertheless, a large part of past research efforts relied on strategies to inhibit upstream components of STAT3 signaling, such as IL-6 or JAKs to treat NSCLC. However, results derived from these studies remain modest [65,93]. Siltuximab (an anti-IL-6 antibody) inhibited growth of xenografted NSCLC cells, but failed to show clinical activity in NSCLC and other solid cancers [95,96]. Also ALD518 (another anti-IL-6 antibody) showed no clinical benefits for NSCLC patients, except for reduction of cancer-associated anemia and cachexia [97,98,99,100]. Although JAK-TKIs performed favorably in preclinical NSCLC models, their enormous success in the treatment of myeloproliferative disorders could not be recapitulated for lung cancer so far [101]. AZD1480 (a JAK1/2-TKI) impaired tumor growth in an EGFR-driven lung cancer mouse model and blocked growth of xenografted solid tumor cells [102,103]. Nonetheless, AZD1480 lacked overall clinical activity in NSCLC and other solid cancers [104]. Similarly, ruxolitinib (another JAK1/2-TKI) had a promising preclinical profile by potently blocking tumor growth in EGFR- and K-RAS-driven lung cancer models and restoring cisplatin-sensitivity to resistant NSCLC cell lines [51,105,106]. However, the combined administration of ruxolitinib and pemetrexed/cisplatin was ineffective for treating NSCLC patients with advanced or recurrent NSCLC. Likewise, ruxolitinib in combination with the first-generation EGFR-TKI erlotinib or the second-generation EGFR-TKI afatinib was overall ineffective in the treatment of TKI resistant *EGFR*-mutated-NSCLCs in clinics. [107,108]. Momelotinib (a JAK1/2 and TBK1-TKI) exhibited similar results. Although momelotinib synergized with a MEK inhibitor and potently induced regression of aggressive *K-RAS-p53* mutated murine lung tumors, the combination was ineffective in patients suffering from refractory, metastatic, *K-RAS*-mutated NSCLC [54,109]. In addition, inhibition of STAT3 signaling in NSCLC is often accompanied by numerous adverse events (AEs). AEs, likely caused as a direct consequence of an anti-IL-6 treatment, include rectal hemorrhage, thrombocyto- and neutropenia, abnormal liver function and infections [96,99,100]. AEs caused by JAK inhibition are relatively similar and include thrombocyto- and neutropenia, abnormal liver function, infections, neurological disorders (e.g., ataxia) and anemia [104,107,108,109]. A potential explanation for occurrence of infections upon JAK1/2-TKI administration and the overall lack of clinical activity of JAK1/2-TKIs might be that JAK1/2 inhibition reduces abundance and function of NK cells, thereby impairing defense against pathogens and anti-tumor immunity [110,111,112]. Given that selective targeting of JAK1 has less detrimental effects on NK cells [113], such agents (AZD4205, itacitinib) might display a more favorable clinical profile. Notably, the combination of osimertinib with AZD4205 or itacitinib is currently under investigation in *EGFR*-mutated NSCLC patients with acquired EGFR-TKI resistance. Both JAK1-TKIs enhanced osimertinib’s potency in *EGFR*-driven NSCLC xenograft models, including therapy resistant EGFR T790M positive ones [114,115]. Clinical trials investigating the combination of STAT3 axis targeting and ICB therapy in NSCLC patients are currently ongoing. A solid preclinical rationale for these clinical trials has been provided by the following outcomes [71,116]: (i) STAT3 regulates PD-L1 expression; and (ii) STAT3 axis targeting improves ICB therapy efficacy in vivo. An overview of the discussed clinical trials is provided in Table 1. 

## 7. STAT3 as a Tumor Suppressor in NSCLC

Although the majority of studies described STAT3 as an oncogene, STAT3 also possesses tumor suppressive activity in a variety of solid cancers, such as brain, colorectal, pancreatic, thyroid, prostate and lung cancer, thereby complicating efforts to successfully implement STAT3 axis targeting into clinics [20,29,117,118,119,120,121,122,123,124,125]. Zhou et al. initially described STAT3 as a tumor suppressor in lung cancer, but also reported opposing roles for STAT3 in tumor initiation and growth in a context-dependent manner [123]. Lung epithelial-specific deletion of STAT3 before urethane-induced cancer formation increased *K-RAS* mutation rates and tumor numbers, while STAT3 deletion after urethane-treatment reduced tumor cell proliferation and tumor growth [123]. Similar effects were observed in a genetically engineered lung cancer mouse model within this study: simultaneous deletion of STAT3 and activation of oncogenic K-RAS in the lung epithelium resulted in increased tumor numbers, but ultimately reduced tumor growth/burden [123]. Altogether, this study elegantly demonstrates the dual role of STAT3 in tumor formation. Under oncogenic stress STAT3 exerts its tumor suppressive function by preventing disease initiation, while in the absence of oncogenic stress it acts as an oncogene and facilitates tumor cell proliferation and tumor growth/burden [123]. Subsequent investigations further confirmed a tumor suppressive role for STAT3 in K-RAS-driven lung tumorigenesis. Grabner et al. showed that lung-tissue specific deletion of STAT3 increases K-RAS-driven tumor initiation and progression in mice [124]. Here, STAT3 regulates NF-κB-dependent expression of tumor cell derived Chemokine (C-X-C motif) ligand 1 (the mouse orthologue to human *IL-8*) by sequestering NF-κB in the cytoplasm, thereby impairing pro-tumorigenic myeloid tumor infiltration and tumor vascularization [124]. However, prominent sex-associated discrepancies have been unraveled by Caetano et al. [125]: lung-epithelial specific deletion of STAT3 diminished *K-RAS*-mutant tumor load in female mice, but increased the tumor burden in male mice. The reason for this is that STAT3 deletion in male mice altered NF-κB activation and *Cxcl1* expression. This led to reprogramming of TME towards a pro-tumorigenic phenotype, ultimately resulting in enhanced tumor formation. Thus, both studies not only illustrate the importance of STAT3-NF-κB-CXCL1 signaling for K-RAS-driven lung tumorigenesis, but also demonstrate that tumor intrinsic STAT3 activation can inhibit the formation of a pro-tumorigenic TME [125]. D’Amico et al. further corroborated a tumor suppressive role for STAT3 in NSCLC and defined STAT3 as an important modulator of epithelial cell identity and differentiation [20]. Subcutaneously transplanted KRAS^G12D^/p53^-/-^ lung epithelial cells without STAT3 showed a faster disease initiation and established tumors that lacked epithelial differentiation, compared to STAT3 proficient cell lines. Thus, STAT3 loss within this model resulted in the formation of a mesenchymal-like tumor phenotype, known to be associated with drug resistance and metastasis. Maintenance of epithelial identity by STAT3 depended on TP63, a gene previously described to be critical for epithelial development in mice and human [20]. Further support for a tumor suppressing function for STAT3 came from analysis of patient cohorts with breast, colorectal and lung cancers: high levels of STAT3 are associated with a good prognosis [124,126,127]. Next to analysis of patient cohorts, experimental studies in other cancers corroborated STAT3‘s role as a tumor suppressor. These studies also indicated that genetic drivers can dictate whether STAT3 acts as a tumor suppressor or oncogene. For instance, subcutaneously injected PTEN-deficient astrocytes without STAT3 formed bigger tumors than STAT3 proficient cells. Conversely, subcutaneously injected EGFR-mutant expressing astrocytes without STAT3 formed no tumors compared to STAT3 proficient cells [117]. Further, Schneller et al. showed that constitutive active STAT3 reduces tumor formation of Ras-transformed, p19^ARF^-deficient hepatocytes, while increasing the tumor formation of Ras-transformed, p19^ARF^-proficient hepatocytes [122].

Collectively, these studies substantiate the previously suggested notion that STAT3 exerts two opposing roles in tumorigenesis depending on the cellular context and/or on the tumor genetic driver [128].

## 8. Conclusions

STAT3 is a critical lynchpin for cytokine, interferon, hormone and growth factor signaling. Under physiological conditions STAT3 is paramount for maintaining the structure and function of the alveolar epithelium [129]. STAT3 activation is frequently found in solid cancers, including NSCLC. In NSCLC patient samples, high pSTAT3 levels correlate with advanced disease stage, smoking and EGFR status and serve as a negative predictor for patient survival. Experimental studies on tumor-cell intrinsic STAT3 signaling mechanisms (particularly in the context of EGFR-driven NSCLC) further consolidated the notion of STAT3 as an oncogene in this disease. In addition, STAT3 was identified to be critically involved in shaping the lung cancer TME by driving tumor promoting inflammation and evasion of anti-tumor immunity. These results have postulated STAT3 signaling as a therapeutic option in solid tumors, including NSCLC. However, blockade of STAT3 signaling may result in an impairment of tumor immune surveillance and STAT3 may act as a tumor suppressor, particularly in the context of K-RAS-driven lung tumorigenesis. These dual STAT3 functions in tumorigenesis need to be taken into account in future clinical trials targeting the STAT3 axis.

## Figures and Tables

**Figure 1 cancers-12-01107-f001:**
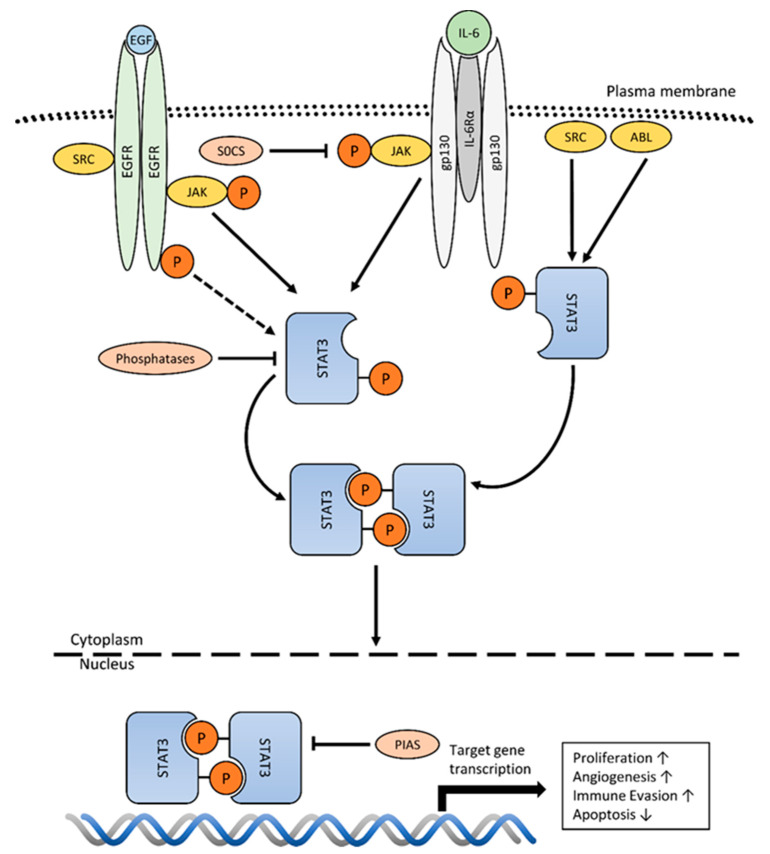
Mechanisms of STAT3 activation: Cytokine binding (e.g., IL-6) to its cognate receptor (e.g., gp130/IL-6Rα) induces receptor dimerization and activation of receptor associated Janus kinases (JAKs). Activated JAKs provide STAT3 receptor-docking sites by phosphorylation of cytoplasmic receptor tails (not shown). Subsequently STAT3 is activated by JAKs due to single tyrosine phosphorylation (Tyr^705^). Formed STAT3-dimers translocate into the nucleus and drive transcription of genes associated with the cancer hallmarks: proliferation, angiogenesis, immune evasion and evasion of apoptosis (middle). Receptors with intrinsic kinase activity (RTKs) like EGFR also facilitate STAT3 activation via JAK engagement rather than directly (left). STAT3 activation has also been reported by non-receptor tyrosine kinases (nRTKs) like SRC or ABL (right). Under physiological conditions, STAT3 activation is tightly controlled by phosphatases, SOCS and PIAS proteins.

**Figure 2 cancers-12-01107-f002:**
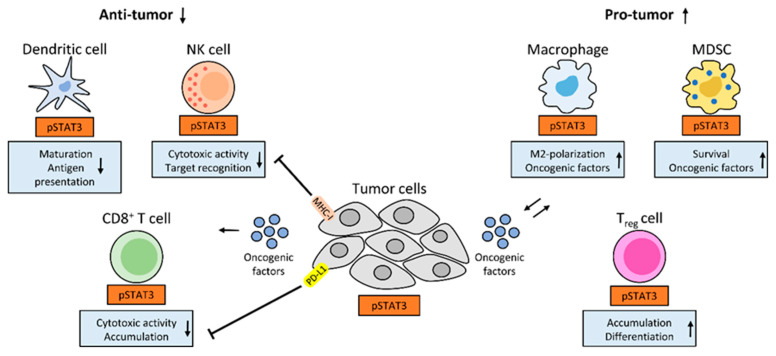
A central role for STAT3 in anti-tumor immunity and inflammation: Tumor cell-intrinsic STAT3 drives the expression of oncogenic factors (e.g., IL-6, IL-10 or VEGF), which can activate STAT3 in immune cells and enhance their recruitment to the TME (middle). STAT3 activity in NK and CD8^+^ T cells impairs their cytotoxic activity, while STAT3 activity in dendritic cells hampers their maturation and capability to present antigens. Tumor cell-intrinsic STAT3 drives MHC class I expression, that impairs cytotoxic activity of NK cells, or drives PD-L1 expression, thereby diminishing cytotoxic activity of CD8^+^ T cells (left). In tumor-infiltrating MDSCs and macrophages, STAT3 drives the expression of oncogenic factors (e.g., VEGF), which amplifies STAT3 activity in tumor cells and leads to tumor growth. Also, survival of MDSCs, M2-macrophage polarization, Treg differentiation and accumulation in the TME are STAT3 dependent (right). STAT3 signaling within tumor cells and immune cells impairs anti-tumor immunity and drives tumor-promoting inflammation.

**Table 1 cancers-12-01107-t001:** Completed and ongoing clinical trials assessing STAT3 pathway inhibition in NSCLC; Source: ClinicalTrials.gov.

Compound (s)	NCT Identifier	Patient Characteristics	Phase	Status/Result
Siltuximab ^a^	NCT00841191	Advanced solid cancers including anti-EGFR therapy resistant NSCLC	Phase 1/2	Completed, no clinical benefit
ALD518 ^a^	NCT00866970	Advanced NSCLC	Phase 2	Completed, reduction of anemia/cachexia
AZD1480 ^b^	NCT01112397	Advanced solid cancers including *EGFR* or *ROS*-mutant NSCLC	Phase 1	Completed, no clinical benefit
Ruxolitinib ^b^ + Pemetrexed/ Cisplatin ^d^	NCT02119650	Advanced or recurrent NSCLC without targetable driver mutations	Phase 2	Completed, no clinical benefit
Ruxolitinib ^b^ + Erlotinib ^d^	NCT02155465	Advanced *EGFR*-mutant + EGFR-TKI resistant (including *EGFR* T790M) NSCLC	Phase 1/2	Completed, no clinical benefit
Ruxolitinib ^b^ + Afatinib ^d^	NCT02145637	Advanced *EGFR*-mutant + EGFR-TKI resistant (including *EGFR* T790M) NSCLC	Phase 1	Completed, no clinical benefit
Momelotinib ^b^ + Trametinib ^d^	NCT02258607	Advanced *K-RAS*-mutated NSCLC with prior failure to platinum-based chemotherapy	Phase 1	Completed, no clinical benefit
AZD4205 ^b^ + Osimertinib ^d^	NCT03450330	Advanced *EGFR*-mutant + EGFR-TKI resistant NSCLC	Phase 1/2	Ongoing, NA
Itacitinib ^b^ + Osimertinib ^d^	NCT02917993	Advanced *EGFR*-mutant+ EGFR-TKI (including *EGFR* T790M) resistant NSCLC	Phase 1/2	Ongoing, NA
Itacitinib ^b^ + Pembrolizumab ^d^	NCT03425006	Advanced PD-L1 expressing NSCLC (first-line treatment)	Phase 2	Ongoing, NA
AZD9150 ^c^+ Durvalumab ^d^	NCT02983578	Advanced solid cancers including treatment-refractory NSCLC	Phase 2	Recruiting, NA
AZD9150 ^c^ + Durvalumab ^d^	NCT03334617	Advanced NSCLC progressed on an anti-PD-1/PD-L1 therapy	Phase 2	Recruiting, NA
AZD9150 ^c^ + Durvalumab ^d,*^	NCT03421353	Advanced solid cancers including treatment refractory NSCLC	Phase 1/2	Ongoing, NA

^a^ Anti-IL-6 antibody: Siltuximab, ALD518; ^b^ JAK-TKIs: AZD1480 (JAK1/2), ruxolitinib (JAK1/2), AZD4205 (JAK1), itacitinib (JAK1), momelotinib (JAK1/2, TBK1); ^c^ AZD9150 (STAT3 antisense oligonucleotide); ^d^ others: pemetrexed/cisplatin (cytostatic), erlotinib (1st generation EGFR-TKI), afatinib (2nd generation EGFR-TKI); osimertinib (3rd generation EGFR-TKI), trametinib (MEK1/2 inhibitor), pembrolizumab (anti-PD-1 monoclonal antibody), durvalumab (anti-PD-L1 monoclonal antibody);* + chemotherapy, NA: not available.

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
