# Peer review of "STAT3: Versatile Functions in Non-Small Cell Lung Cancer"

_cancers, 2020, doi:10.3390/cancers12051107_

Round 1
Reviewer 1 Report
This is a well-written and easily comprehensible review paper. Despite there have been several STAT-3 review articles published in the Cancers journal in these two years, this review paper further provides concise but important contents describing the functions of STAT-3 in the stromal cells of tumor microenvironment and updating the status of the clinical trials targeting STAT-3 pathway. This reviewer would like to raise the following two comments for making the paper much better:
(1) For the development and progression of NSCLC, the roles of STAT-3 in the stromal cells should be as critical as those in cancer cells. The information for Section 5 should be weighted and an extra Figure 2 are strongly suggested.
(2) Given STAT-3 is a key protein in diverse cell types and exhibits versatile functions in the tumor microenvironment, the paper will be advanced if the authors could consider and discuss the side-response(s) of NSCLC patients in the clinical trials summarized in Table 1 and Section 6.
Author Response
Reviewer 1
- This is a well-written and easily comprehensible review paper. Despite there have been several STAT-3 review articles published in the Cancers journal in these two years, this review paper further provides concise but important contents describing the functions of STAT-3 in the stromal cells of tumor microenvironment and updating the status of the clinical trials targeting STAT-3 pathway.
We are glad that the work is well received and want to thank the reviewer for this positive evaluation at this point.
For the development and progression of NSCLC, the roles of STAT-3 in the stromal cells should be as critical as those in cancer cells. The information for Section 5 should be weighted and an extra Figure 2 are strongly suggested.
The reviewer is absolutely right. For NSCLC tumorigenesis STAT3 activation within stromal cells is as critical as its activation within cancer cells. To highlight this fact, we included now an extra figure in the manuscript (Figure 2).
Page: 5/6 Line: 225-245
Given STAT-3 is a key protein in diverse cell types and exhibits versatile functions in the tumor microenvironment, the paper will be advanced if the authors could consider and discuss the side-response(s) of NSCLC patients in the clinical trials summarized in Table 1 and Section 6.
We agree with the reviewer; this is an important point to consider. STAT3 is ubiquitously expressed and intact STAT3-signaling is crucial to maintain cellular homeostasis in numerous cell types. Hence, STAT3-targeting in NSCLC and other malignancies is unfortunately often accompanied by various adverse events (e.g.; neutropenia, impairment of liver function, opportunistic infections, etc.). A paragraph outlining frequent drug-related adverse events is now included in the manuscript.
Page: 7 Line: 275-283

Reviewer 2 Report
Τhis is a very well organized, structured and executed comprehensive review on the binary role of STAT3-mediated signaling in NSCLC carcinogenesis and expansion. The coverage of the literature is thorough, critical and concise.
The authors are to be congratulated for the work. I have only one minor suggestion to make: Given the documented binary role of STAT3 as both a tumor activating pathway and a tumor suppressor depending on the context and tumor microenvironment, the authors could devote a paragraph elaborating on the cellular context and possible genetic drivers that determine whether STAT3 will act as a photo-oncogene or a tumor suppressor gene. Are there any literature data on this?
Otherwise the review is ready for publication in its current form
Author Response
Reviewer 2
- This is a very well organized, structured and executed comprehensive review on the binary role of STAT3-mediated signaling in NSCLC carcinogenesis and expansion. The coverage of the literature is thorough, critical and concise. The authors are to be congratulated for the work.
We are glad that the reviewer appreciates our manuscript and we want to express our gratitude at this point.
- Given the documented binary role of STAT3 as both a tumor activating pathway and a tumor suppressor depending on the context and tumor microenvironment, the authors could devote a paragraph elaborating on the cellular context and possible genetic drivers that determine whether STAT3 will act as a photo-oncogene or a tumor suppressor gene. Are there any literature data on this? Otherwise the review is ready for publication in its current form.
As pointed out, STAT3 can act as tumor suppressor especially in the context of K-RAS driven lung tumorigenesis, while STAT3 acts rather as an oncogene in the context of EGFR-driven lung tumorigenesis. This observation indicates that cancer cell genetic drivers dictates whether STAT3 acts as a tumor suppressor or oncogene. Studies by De la Iglesia et al. and Schneller et al. support this idea. Indeed, De la Iglesia et al. showed that subcutaneously transplanted PTEN-deficient astrocytes without STAT3, formed bigger tumors than STAT3 proficient cells, while conversely subcutaneously transplanted EGFR-mutant expressing astrocytes lacking STAT3, formed smaller tumors than STAT3 proficient cells. In addition, Schneller et al. showed that STAT3 reduces tumor formation of Ras-transformed, p19ARF-deficient hepatocytes, while increasing tumor formation of Ras-transformed, p19ARF-proficient hepatocytes. Results of the studies are now incorporated in the manuscript.
Page: 8/9 Line: 337-344
Ref: De la Iglesia, N.; Konopka, G.; Puram, S.V.; Chan, J.A.; Bachoo, R.M.; You, M.J.; Levy, D.E.; Depinho, R.A.; Bonni, A. Identification of a PTEN-regulated STAT3 brain tumor suppressor pathway. Genes Dev 2008, 22, 449-462, doi:10.1101/gad.1606508.
Ref: Schneller, D.; Machat, G.; Sousek, A.; Proell, V.; van Zijl, F.; Zulehner, G.; Huber, H.; Mair, M.; Muellner, M.K.; Nijman, S.M., et al. p19(ARF) /p14(ARF) controls oncogenic functions of signal transducer and activator of transcription 3 in hepatocellular carcinoma. Hepatology 2011, 54, 164-172, doi:10.1002/hep.24329.
